# Host-Affected Body Coloration Dynamics in *Perina nuda* Larvae: A Quantitative Analysis of Color Variations and Endogenous Plant Influences

**DOI:** 10.3390/insects16070728

**Published:** 2025-07-17

**Authors:** Songkai Liao, Xinjie Mao, Yuan Liu, Guihua Luo, Jiajin Wang, Haoyu Lin, Ming Tang, Hui Chen

**Affiliations:** 1State Key Laboratory of Conservation and Utilization of Subtropical Agro-Bioresources, Guangdong Laboratory for Lingnan Modern Agriculture, College of Forestry and Landscape Architecture, South China Agricultural University, Guangzhou 510642, China; liaosongkai2023@163.com (S.L.); mxj20128@163.com (X.M.); 18203927730@163.com (Y.L.); lgh1653007221@163.com (G.L.); a947375051@stu.scau.edu.cn (J.W.); tangming@scau.edu.cn (M.T.); 2Fujian Academy of Forestry, Fuzhou 350012, China; jolhy@163.com

**Keywords:** *Perina nuda*, larva, body coloration, host, quantitative analysis, computer image analysis

## Abstract

*Perina nuda* larvae are serious pests of banyan trees in southern China, causing significant damage. This study used computer technology to accurately measure color changes in different parts of the larvae as they fed on various banyan species. Additionally, we analyzed the chlorophyll, carotenoid, soluble sugar, soluble protein, tannin, flavonoid, and total phenol contents in the leaves to understand their effects on larval coloration. Results showed that chlorophyll-b in the leaves had the most significant impact on the body coloration of larvae, making certain body parts lighter or darker. Flavonoids also influenced coloration but to a lesser extent. Understanding how host plant chemicals influence larval coloration may provide insights that could inform the development of targeted pest management strategies, thereby contributing to the health and sustainability of banyan trees and their ecosystems.

## 1. Introduction

Color is a complex trait determined not only by genetic factors but also by environmental influences on the phenotype [1]. The RGB color space is a system of adding light, in which R represents red, G represents green, and B represents blue. By combining different RGB values, almost all perceivable colors can be created [2]. The R, G, and B values range from 0 to 255, allowing the RGB space to represent 256 × 256 × 256 colors, totaling 16,800,000, commonly referred to as 16 million or 10 million colors [3]. Lightness (L) indicates the degree of light and dark in a color [4].

Body coloration in insects is an important ecological trait influencing their survival and reproduction [5,6,7]. Insect body coloration primarily serves protective and recognition functions [8,9]. Insects achieve protection by mimicking the predominant habitat colors to evade predators, as seen in stick insects in forests, green mantises in grasslands, and geometrid moths on tree trunks [6]. The recognition function is evident in intraspecific insect communication, where coloration and patterns aid in identifying conspecifics, especially during courtship and competition [10]. Additionally, insect body coloration can regulate body temperature by absorbing light [11]. Body coloration significantly impacts insect adaptability through dual selective pressures: predator avoidance (particularly against avian UV vision) and physiological optimization (thermoregulation, UV resistance, and immune function) [6].

The diversity of insect body coloration is gradually formed under long-term natural selection [12,13,14]. The coloration of insects can be classified into pigmentary coloration and structural coloration types [15]. Pigmentary coloration arises from physiological metabolism, where metabolites are deposited in the insect cuticle or epidermis, absorbing certain wavelengths of light and reflecting other visible light, thereby forming specific colors. Structural coloration results from the ultrastructural features of insect cuticles, such as scales or melanin granules, causing light diffraction, interference, or refraction. The synthesis and metabolism of pigments within insects are important material bases for the diverse body coloration patterns observed. Among these, ommochromes, melanin, and pteridines are the most important pigments influencing insect coloration [16,17,18]. During growth, insects like the silkworm utilize uric acid, carotenoids, and flavonoid pigments, along with the three primary substances, enhancing their coloration diversity [19].

At present, there are three theories on the mechanism of insect body coloration differentiation: the control theory of environmental factors, the regulation theory of insect hormones, and the theory of genetic factors [20,21,22,23,24]. Across all three mechanisms, environment-mediated changes in body coloration require insects to sense environmental signals in advance and induce phenotype changes to tackle future challenges [25,26]. The body coloration of some insects is affected by external factors such as host plants, population density, temperature, humidity, and background color. For example, the elytral color of *Dynastes hercules* Linnaeus beetle responds to air humidity signals, appearing brownish yellow in dry conditions and gradually darkening with increasing humidity [27]. The larvae of *Biston betularia* L. mimic the color of their surroundings, like tree branches [28]. *Helicoverpa armigera* Hübner larvae show more green when fed on leaves compared to flowers and fruit [29]. However, previous studies on insect body coloration usually adopted qualitative methods. Different observers have different concepts of color, and this kind of color classification process will produce certain errors depending on the subjective consciousness of observers and the surrounding environmental factors. Li et al. [30] used computer image analysis to quantify the *Prodenia litura* Fabricius larval body coloration, assessing age and light intensity effects. Peng et al. [31] analyzed the R, G, B, and L values of different parts of *P. litura* larvae fed on differently colored diets, scientifically unveiling coloration change patterns.

In this study, we focus on the insect *Perina nuda* Fabricius (Lepidoptera: Lymantriidae), a prominent defoliator of forests in Southeast Asia and the southern regions of East Asia, which is a significant threat to banyan trees (*Ficus* spp.), making it a serious pest [32,33,34]. The head of the *P. nuda* larvae is gray, the central dorsal thorax and abdomen is earthy yellow, the back of the fourth and fifth somite has an egagropilus in brownish black, and the sixth to ninth somite can be seen in the center of the back with the obvious yellow double stripes; both sides of each somite is red (Figure 1). The first- to third-instar larvae are light in body coloration, and the overall color is gradually deepened with age.

The coloration of leaves is a complex process influenced by factors like the pigment content, types of pigments, nutrients, and phenolic compounds [35]. The main pigments in leaves are chlorophyll and carotenoids. Chlorophyll, comprising chlorophyll-a and chlorophyll-b, gives leaves their green color, while carotenoids interact with chlorophyll to produce yellow to orange hues [36]. The influence of soluble sugars and soluble proteins on leaf color is indirect [37]. Sugars can combine with anthocyanins to form anthocyanidins, which contribute to leaf reddening [38,39,40]. Stress on plants leads to reactive oxygen species accumulation, reducing soluble proteins and damaging lipid and protein structures, making chlorophyll prone to degradation, which affects leaf color [41]. The accumulation of phenolic compounds in plants is associated with darker colors [42,43,44]. High concentrations of tannins in plant leaves can inhibit the degradation of anthocyanins mediated by peroxidase [45]. Among flavonoid substances, anthocyanins form pigment complexes with flavones, flavonols, and other auxiliary pigments, giving plants red, blue, purple, yellow, and other colors [46].

Despite many studies on factors affecting body coloration in lepidopteran species [47,48,49,50,51,52], little is known about those involved in larval body coloration of *P. nuda*. In this study, we selected four host plants on which to rear larvae. We digitized the color indices (R, G, B, and L values) of various regions of the larvae (including the head, dorsal thorax and abdomen, stripe, dorsal mid-line, and tail) that fed on four hosts in order to determine the patterns of changes in body coloration. Also, we digitized the color indices and determined the content of endogenous substances (chlorophyll, carotenoid, soluble sugar, soluble protein, tannin, flavonoid, and total phenol) of host leaves which were fed to 20-day-old larvae. Redundancy analysis (RDA) illustrated the influence of these substances on larval coloration, leading to a comprehensive model delineating the relationship between host leaves and larval color. These approaches aimed to establish a more precise relationship compared to visual observation alone, thereby exploring the impact of host plants on larval body coloration. Ultimately, our study provides a foundational framework for future research into insect body coloration.

## 2. Materials and Methods

### 2.1. Test Plants

Four species of *Ficus* which are the most widely distributed in Guangdong province were selected: *F. microcarpa* L., *F. altissima* Blume, *F. concinna* Miquel, and *F. benjamina* L. The leaves of different host plants used in the experiment were healthy with the same growth and collected from the campus of South China Agricultural University, Guangzhou, China.

### 2.2. Origin of Insects

The *P. nuda* colony used in this study originated from a field-collected adult on the campus of South China Agricultural University, Guangzhou, China, in 2021, and has been subsequently maintained as a laboratory population through multigenerational rearing on mixed foliage of the above four *Ficus* species. In field observation and investigation, it was observed that *P. nuda* undergoes 8–9 generations annually in Guangzhou, with substantial generational overlap and the absence of a distinct overwintering generation, and the larvae of *P. nuda* exhibited 6–8 larval instars per generation. A pair of adults was transferred to a mating jar, and offered paper strips for egg-laying. The substrate was changed on alternate days (fresh leaves were provided). All experiments were conducted in an incubator (MGC-450HP, Shanghai Yiheng Technology Instrument Co., Ltd., Shanghai, China) at 24 ± 1 °C, 40% relative humidity, and a photoperiod of 12:12 (L:D) h.

### 2.3. Chlorophyll and Carotenoid Content of Leaves

Ground leaf powders from samples fed to 20-day-old larvae were extracted using 80% acetone and centrifuged, with light absorbance recorded at wavelengths corresponding to chlorophyll-a, chlorophyll-b, and carotenoids using a UV-Vis spectrophotometer [45]. The total chlorophyll content was the sum of the contents of chlorophyll-a and chlorophyll-b. Five samples were taken for content determination for each substrate, with each sample measured in triplicate.

### 2.4. Nutrient and Phenolic Compound Content of Leaves

Different host plant leaves for 20-day-old larvae feeding were washed with deionized water, cut into small pieces, and divided for further measuring. Soluble sugars and soluble proteins, as well as three phenolic compounds—tannins, flavonoids, and total phenols—were determined using the Plant Soluble Sugar Content Assay Kit (KT-2-Y, Suzhou Comin Biotechnology Co., Ltd., Suzhou, China), BCA Protein Content Assay Kit (BCAP-2-W, Suzhou Comin Biotechnology Co., Ltd., Suzhou, China), Tannin Content Assay Kit (DN-2-Y, Suzhou Comin Biotechnology Co., Ltd., Suzhou, China), Plant Flavonoid Test Kit (LHT-2-G, Suzhou Comin Biotechnology Co., Ltd., Suzhou, China), and Plant Total Phenol Test Kit (TP-1-G, Suzhou Comin Biotechnology Co., Ltd., Suzhou, China), respectively. Five samples were taken for content determination for each substance, with each sample measured in triplicate.

### 2.5. Insect and Leaf Color

To standardize leaf and larval body color measurements, digital photographs were taken under controlled lighting conditions. Images were captured using a SONY ILCE-7 camera (SONY, Minato City, Japan) with predetermined settings (macro mode, automatic white balance, no flash) in a photo studio setup. The shooting background was 18% neutral gray cardboard, which could reflect the average brightness of the real scene; the amount of exposure of 18% gray cardboard aligns with the optimum amount of exposure [53]. The 20-day-old larvae or the leaves for their feeding were placed directly below the camera on the cardboard background and photographs were taken once they became still. The R, G, and B color values of leaf surfaces and larval body regions (head, dorsal thorax and abdomen, stripe, dorsal mid-line, and tail; Figure 1) were recorded using Instant Color Picker software (v2.0.5.32), and L values were calculated to account for the varying sensitivities of the human eye to red, green, and blue color channels using the following formula:(1)L=0.299∗R+0.587∗G+0.114∗B

### 2.6. Statistical Analysis

The R, G, B, and L values of different host leaves are named Rh, Gh, Bh, and Lh, respectively. The R, G, B, and L values of the body coloration are named Rb, Gb, Bb, and Lb, respectively. Nonlinear regression models were constructed in SPSS (v26) to characterize the relationships between L value of different body regions of *Perina nuda* larvae across different developmental ages and host plant leaves. Model visualizations were generated using GraphPad Prism (v9.5). A three-way ANOVA was implemented to evaluate the main effects and interactions of the host plant, developmental age, and body region on larval L values, with post hoc Tukey’s HSD tests (usually <0.05) applied for pairwise comparisons of L values between host treatments within matched developmental ages and body regions. Redundancy analysis (RDA) was performed using Canoco (v5.0) to quantify the contribution of host endogenous biochemical components to the variation in body coloration parameters (Rb, Gb, Bb) across distinct anatomical regions of 20-day-old larvae. Linear regression models were further established in SPSS (v26) to assess the associations between larval body coloration (dependent variables: Rb, Gb, Bb) and host leaf chromatic traits (independent variables: Rh, Gh, Bh) at 20 days old, with regression diagnostics and graphical representations generated in GraphPad Prism (v9.5).

## 3. Results

### 3.1. Changing Trend of the Body Coloration of P. nuda Larvae in Different Ages

As illustrated in Figure 2, the trends in the L (lightness) value of different body regions of *Perina nuda* larvae fed on different host plants exhibited analogous patterns among hosts. The following results were obtained under all four host conditions: (1) Head: The L values were the highest for newly hatched larvae (1-day-old, followed by a rapid decline, a gradual increase, and eventual stabilization over time). (2) Dorsal thorax and abdomen: The L value peaked at 1-day old, showing an initial sharp decrease before transitioning to a slower decline and stabilization. (3) Stripe: The L value was highest at 1-day old and exhibited a gradual decline throughout the developmental period. (4) Dorsal mid-line: The L value displayed a steady increase followed by stabilization. (5) Tail: The L value was highest at 1-day old, followed by a rapid decline, stabilization, and a subsequent slow decrease. These coordinated ontogenetic shifts in lightness reflect conserved temporal patterning across different host treatments, notwithstanding variations in absolute chromatic parameters.

### 3.2. Effects of Different Hosts on Body Coloration of Different Parts of P. nuda Larvae

The L value varied significantly with age, host, and body region (Table 1). In addition, all of the interactions among the three factors were found to be significant (Table 1). The L values for various body regions of larvae at different developmental stages and upon feeding on different *Ficus* species are shown in Figure 3.

#### 3.2.1. Head

For the head, 5-day-old larvae fed on *F. benjamina* had significantly higher L values than those fed on *F. macrocarpa*. Nine-day-old larvae fed on *F. macrocarpa* had significantly lower L values than those fed on the other three species. Thirteen-day-old larvae fed on *F. benjamina* had significantly higher L value than those fed on *F. macrocarpa* and *F. altissima*, while thirteen-day-old larvae fed on *F. concinna* also had significantly higher L values than those fed on *F. macrocarpa*. Seventeen-day-old larvae fed on *F. macrocarpa* had significantly higher L values than those fed on the other three species. Twenty-one-day-old larvae fed on *F. macrocarpa* had significantly higher L values than those fed on *F. concinna* and *F. benjamina*. Additionally, 21-day-old larvae fed on *F. altissima* had significantly higher L values than those fed on *F. concinna*.

#### 3.2.2. Dorsal Thorax and Abdomen

For the dorsal thorax and abdomen, 9-day-old larvae fed on *F. altissima* had significantly lower L values than those fed on the other three species. Thirteen-day-old larvae fed on *F. benjamina* had significantly higher L values than those fed on *F. altissima* and *F. concinna.* Additionally, thirteen-day-old larvae fed on both *F. macrocarpa* and *F. concinna* had significantly higher L values than those fed on *F. altissima.* Seventeen-day-old larvae fed on *F. macrocarpa* had significantly higher L values than those fed on the other three species, and twenty-one-day-old larvae fed on *F. macrocarpa* had significantly higher L values than those fed on *F. altissima* and *F. concinna.*

#### 3.2.3. Stripe

For the stripe, 9-day-old larvae fed on *F. benjamina* had significantly higher L values than those fed on *F. altissima* and *F. concinna.* Thirteen-day-old larvae fed on *F. macrocarpa* had significantly lower L values than those fed on the other three species. Seventeen-day-old larvae fed on both *F. altissima* and *F. benjamina* had significantly higher L values than those fed on *F. macrocarpa.*

#### 3.2.4. Dorsal Mid-Line

For the dorsal mid-line, 5-day-old and 9-day-old larvae fed on *F. macrocarpa* had significantly higher L values than those fed on the other three species. Additionally, 5-day-old and 9-day-old larvae fed on both *F. concinna* and *F. benjamina* had significantly higher L values than those fed on *F. altissima.* Thirteen-day-old larvae fed on *F. altissima* had significantly lower L values than those fed on the other three species. Seventeen-day-old larvae fed on *F. benjamina* had significantly higher L values than those fed on *F. macrocarpa* and *F. concinna*, and seventeen-day-old larvae fed on *F. altissima* had significantly higher L values than those fed on *F. macrocarpa.*

#### 3.2.5. Tail

For the tail, 5-day-old larvae fed on *F. concinna* had significantly higher L values than those fed on *F. macrocarpa* and *F. altissima.* Five-day-old larvae fed on *F. benjamina* had significantly higher L values than those fed on *F. altissima.* Nine-day-old larvae fed on *F. altissima* had significantly lower L values than those fed on the other three species. Thirteen-day-old larvae fed on both *F. concinna* and *F. benjamina* had significantly higher L values than those fed on *F. macrocarpa* and *F. altissima.* Seventeen-day-old larvae fed on *F. benjamina* had significantly higher L values than those fed on the other three species, while seventeen-day-old larvae fed on *F. macrocarpa* had significantly lower L values than those fed on *F. altissima* and *F. concinna.*

These consolidated results highlight significant variations in L value across different body regions, developmental ages, and *Ficus* species consumed, providing a comprehensive overview of the observed trends.

### 3.3. Correlation Between Body Coloration of 20-Day-Old Perina Nuda Larvae and Endogenous Substances in Host Plants

Detrended correspondence analysis (DCA) was initially performed to assess the gradient length and underlying data structure of the body coloration indices of *P. nuda* larvae in relation to host leaf endogenous substances. The DCA results revealed short gradient lengths across all axes (<3; Appendix A), indicating that the data exhibit linear relationships rather than unimodal distributions. This warranted the selection of redundancy analysis (RDA). Chlorophyll-b (CB) demonstrated the highest explanatory contribution (33.2%), validated through Monte Carlo permutation tests (pseudo-F = 4.2; *p* = 0.008), indicating statistically significant effects on body coloration variation in *P. nuda* larvae. Flavonoids (FL: 15.9%; pseudo-F = 2.1; *p* = 0.086) and chlorophyll-a (CA: 14.3%; pseudo-F = 2.1; *p* = 0.074) exhibited subthreshold potentiality (*p* < 0.10), suggesting marginal directional effects on pigmentation dynamics. Non-significant contributions (*p* > 0.05) were observed for soluble proteins (SP, 9.2%), tannins (TA, 8.7%), and carotenoids (CR, 6.4%); each contribution to the total variance was lower than 10% (Table 2). The RDA (Figure 4) revealed significant spatial correspondence between host leaf endogenous substances and body coloration indices in *P. nuda* larvae. Among the substances evaluated, CB and FL displayed the strongest significant effect, contrasting with weak covariation for soluble sugar (SS) and total phenolic (TP). Strong negative correlations were observed between CB and the R/G/B values of the dorsal thorax and abdomen (R2/G2/B2), suggesting that CB may significantly influence the dorsal thorax and abdomen coloration of *P. nuda* larvae. Though moderately weaker, CA demonstrated negative associations with R2/G2/B2, indicating that CA plays a potential regulatory role in dorsal thorax and abdomen color formation. Conversely, FL exhibited robust positive correlations with the R/G/B values of the dorsal mid-line (R4/G4/B4), indicating its potential to regulate the coloration of the dorsal mid-line of *P. nuda* larvae. Furthermore, the positive effects of CB on R4/G4 indicate potential co-regulatory mechanisms between CB and FL in modulating dorsal mid-line coloration dynamics.

### 3.4. Linear Relationship Between Leaf Color and Body Coloration of 20-Day-Old P. nuda Larvae

Weak but statistically significant associations (*p* < 0.05; R^2^ = 0.117–0.154) were detected between host leaf color and body coloration (Table 3). Specifically, positive correlations emerged between the R/G values of host leaf color (Rh/Gh) and the R value of body coloration (Rb) of the head. Conversely, a negative relationship was observed for Rh relative to the B value of body coloration (Bb) of the tail. In contrast, regression models for the dorsal thorax and abdomen and dorsal mid-line coloration exhibited negligible explanatory power (R^2^ ≤ 0.059, *p* > 0.05), suggesting that host leaf coloration lacks significant predictive capacity for the body coloration of *P. nuda*.

## 4. Discussion

Our results demonstrate that the larval body coloration of *P. nuda* is significantly affected by the *Ficus* species consumed, developmental age, and body region examined. Notably, the lightness varied significantly among larvae fed on different host plants, with distinct patterns observed in the head, dorsal thorax and abdomen, stripe, dorsal mid-line, and tail. However, the trends in the L value of different body regions of *P. nuda* larvae fed on different host plants exhibited analogous patterns, indicating that host choice does not change the overall coloration trend. These findings underscore the importance of host plant selection in larval pigmentation, which may have adaptive significance. The significant variation in L values across different body regions suggests that pigmentation is not uniformly regulated throughout the larva. For instance, larvae fed on *F. benjamina* generally exhibited higher L values in certain body regions at specific developmental ages compared to those fed on other *Ficus* species. This indicates that the pigment deposition or structural coloration mechanisms may differ among body regions, potentially due to differential expression of pigmentation genes or local pigment accumulation.

Our redundancy analysis (RDA) revealed that chlorophyll-b (CB) in host leaves was the top contributor (33.2%) to the variation in larval body coloration, particularly influencing the R, G, and B values of the dorsal thorax and abdomen. This strong negative correlation suggests that higher CB content leads to lighter pigmentation in larvae. Chlorophyll-a (CA) showed the same effects on larval coloration. Chlorophyll is an essential component of herbivores’ diet [54]. Chlorophyll is classified into chlorophyll-a and chlorophyll-b, which exhibit blue-green and yellow-green coloration, respectively, in plant leaves [55]. Though no existing study directly links chlorophyll to insect coloration, it might indirectly affect *P. nuda* larvae. Possible degradation of Chl a/b to pheophorbides and pyropheophorbides might occur in the gut, while still retaining the fundamental porphyrin ring structure [54,56]. These degradation products retain a structural similarity to chlorophyll, enabling them to absorb and reflect specific wavelengths of light, thereby exhibiting shades of green, brown, or darker tones. The accumulation of these pigments in the digestive tract of *P. nuda* larvae may exert a direct influence on their body coloration. Phenolic compounds also play a role in influencing the development of insects [57,58,59]. Although tannins and flavonoids both belong to the class of phenolic compounds, our study revealed only a positive correlation between FL and the R, G, and B values of the dorsal mid-line, which indicates that flavonoids may contribute to darker pigmentation. The FL in the insect body, acquired through feeding on plants [60], may be metabolized and integrated into the larval cuticle or epidermis, affecting coloration. Nutrients such as soluble sugars and soluble proteins exhibited negligible effects on the body coloration of *P. nuda*, suggesting that these nutrients are primarily utilized for growth and development in insects, rather than playing a significant role in the formation of body coloration [61]. Linear regression between leaf color and larval coloration of *P. nuda* indicated weak but significant associations in areas like the head and tail, suggesting biochemistry over direct leaf color impact. Overall, host leaf color has limited influence on larval pigmentation, pointing to a complex interplay of multiple plant factors.

Body coloration change is a common animal adaption to the environment, involving pigment cell reorganization and dispersion. This adaptation can serve various purposes such as self-protection and mate selection [62]. Research has shown that darkened butterflies and moths can rapidly absorb and dissipate heat through their darkened body surfaces to regulate body temperature [63]. Additionally, insects with darker body colorations are often more camouflaged, reducing risk from predators (such as birds) [64,65,66], and tend to show stronger vitality and better evasion capabilities [17]. Mimicry is widespread in insects and can occur in all life stages. Diet is considered one of the most unpredictable environmental factors affecting genetic variation in butterflies [67]. Mimicking plants is significant for both herbivorous insects to evade predators and for carnivorous insects in hunting [68]. In our experiment, we found that the body coloration of *P. nuda* larvae is influenced by host plants, with the L values of most body regions highest at hatching but decreasing as they develop. This change suggests that the process of feeding on *Ficus* enhances larval camouflage, allowing them to better match the background color of leaves and branches. Based on the equations constructed from host leaf color values affecting the body coloration of *P. nuda*, it is evident that the influence of R/G values of leaves on the body color values is greater than that of B values. Notably, the G values of leaves significantly positively impacts the R/G values of body coloration of the larvae’s head, suggesting that higher G values may enhance pigment expression in larvae, thereby aiding in better camouflage against green leaf backgrounds. Conversely, the R value of leaves negatively impacts the B value of body coloration of the larvae’s tail, while the G value of leaves exhibits a positive effect. This finding implies that the *P. nuda* larvae may finely tune their body coloration to enhance visual consistency with the host environment, thereby reducing predation risk. This could be one of the reasons for mimicry in herbivorous insects, providing a basis for future research on the causes of mimicry. Additionally, brightly colored ornamental insects are highly favored by people. There is great potential for using diet components and colors to change the body coloration of insects in the breeding of ornamental insects [31,69].

This experiment investigated the influence of host plants on the body coloration of *P. nuda* larvae under consistent growth conditions. It departed from previous subjective qualitative analyses of insect body coloration, instead employing computer image analysis for quantitative analysis, which more scientifically and intuitively revealed the patterns of insect body coloration change. Unlike most studies using artificial diets, this experiment used natural host plant leaves as food for *P. nuda* larvae. The differences in larval feeding on different hosts could affect the absorption of nutrients, secondary metabolites, and pigments in host plant leaves, thereby affecting body coloration change. Liao et al. [32] found that in non-choice tests, fifth-instar larvae did not show a significant preference for any of the four species of banyan trees, while sixth-instar larvae exhibited a preference for *F. concinna*. In our experiment, the relative body coloration values of *P. nuda* larvae fed on *F. concinna* were not significantly higher or lower in most age stages. This may be due to the small differences in food intake of *P. nuda* larvae on the four host species from 1 to 21 days old, which needs further confirmation through measurements of larval food intake at each age stage. While our study provides valuable insights, there are limitations that warrant consideration. The biochemical analyses focused on select endogenous substances in the host plants, potentially missing other influential compounds. Additionally, although we established correlations between host plant components and larval coloration, the causal mechanisms remain to be elucidated. Future studies involving manipulative experiments, such as feeding larvae with purified compounds or employing RNA interference targeting pigmentation pathways, could provide deeper insights into the underlying processes.

## 5. Conclusions

This study provides critical insights into the relationship between *Perina nuda* larval body coloration and the endogenous substances and color index of their host plants. Using quantitative techniques, we found that the body coloration of larvae varies significantly depending on the host banyan species, their developmental stage, and the specific body regions. Chlorophyll-b was identified as the primary factor influencing larval coloration, particularly affecting the dorsal thorax and abdomen regions. Flavonoids also showed a noticeable, though lower, impact by influencing the dorsal mid-line coloration. Our findings suggest that the chemical properties of host plants, rather than just visual characteristics, play a significant role in shaping the adaptive pigmentation of these larvae. Our work lays the groundwork for more detailed studies into the ecological and biochemical mechanisms that drive insect pigmentation and adaptation.

## Figures and Tables

**Figure 1 insects-16-00728-f001:**
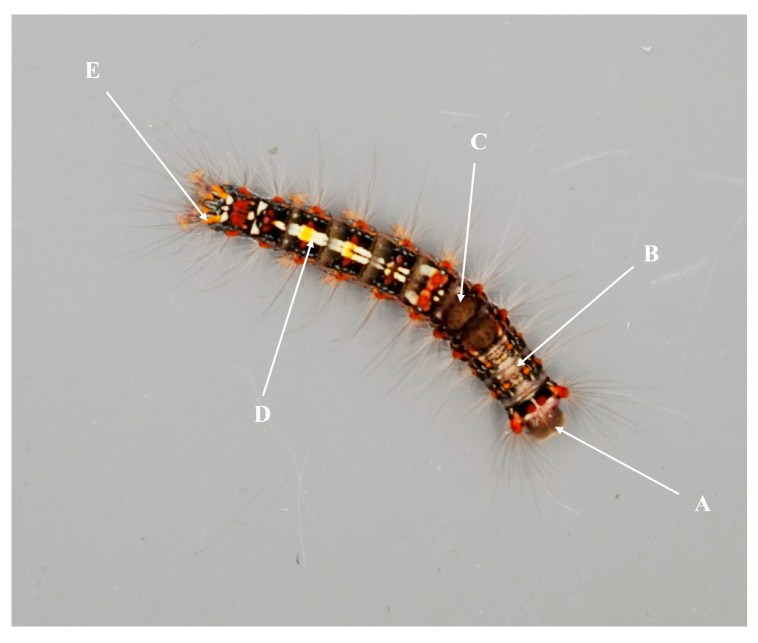
Color sampling position of *Perina nuda* larva: (A) head, (B) dorsal thorax and abdomen, (C) stripe, (D) dorsal mid-line, (E) tail.

**Figure 2 insects-16-00728-f002:**
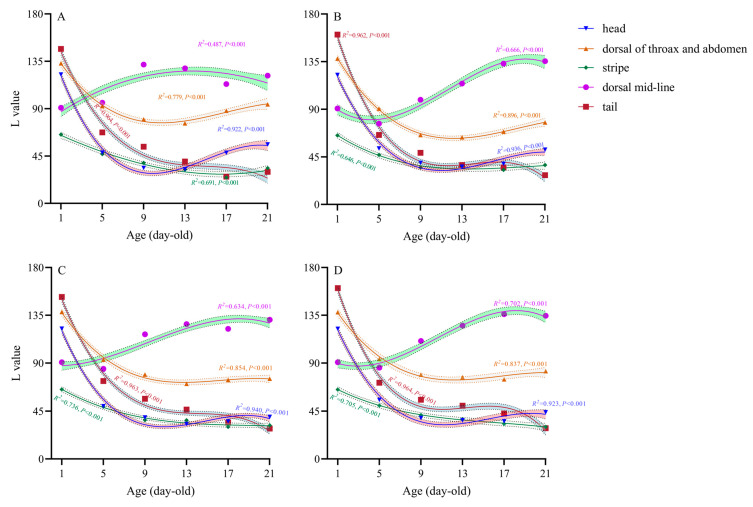
Trends in L (lightness) values of *Perina nuda* larval body coloration across different developmental ages when reared on various host plants. (**A**) Larvae fed on *Ficus macrocarpa*. (**B**) Larvae fed on *Ficus altissima*. (**C**) Larvae fed on *Ficus concinna*. (**D**) Larvae fed on *Ficus benjamina*.

**Figure 3 insects-16-00728-f003:**
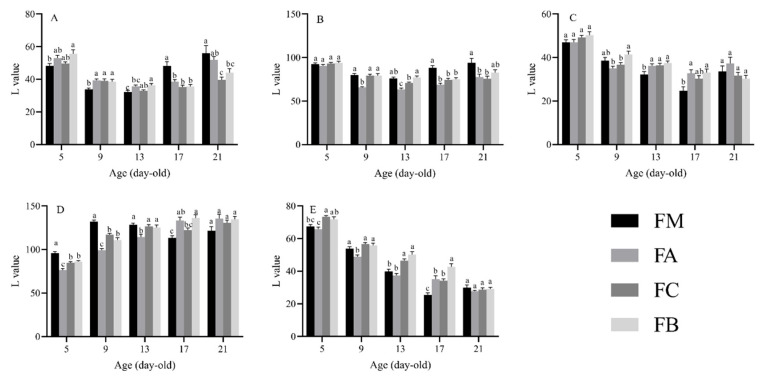
The L (lightness) value of *Perina nuda* larvae fed on different hosts. Different lowercase letters on the column indicated significant differences in different banyan species (*p* < 0.05), as determined by Tukey’s HSD post hoc comparisons. (**A**) Head. (**B**) Dorsal thorax and abdomen. (**C**) Stripe. (**D**) Dorsal mid-line. (**E**) Tail. (FM) Larvae fed on *Ficus macrocarpa*. (FA) Larvae fed on *Ficus altissima*. (FC) Larvae fed on *Ficus concinna*. (FB) Larvae fed on *Ficus benjamina*.

**Figure 4 insects-16-00728-f004:**
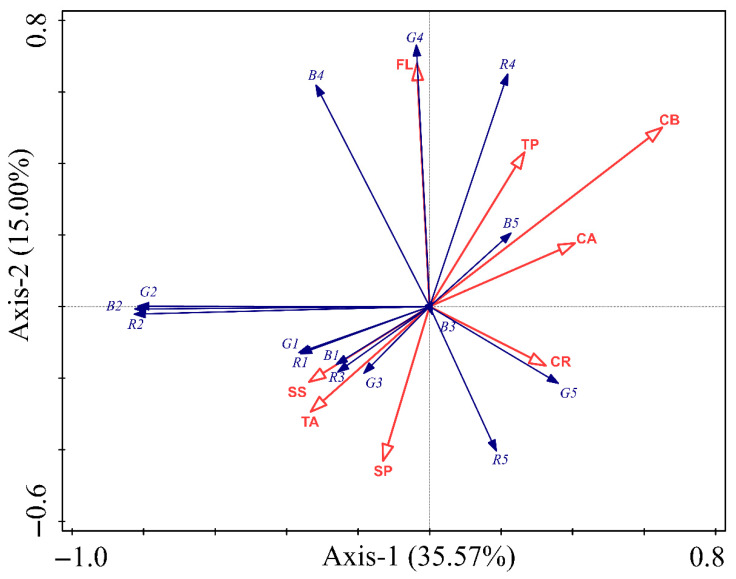
RDA ordination map of endogenous substances of host leaves and body color indices of *Peina nuda* larvae. Red arrows signify various endogenous substances, including soluble sugar (SS), soluble protein (SP), tannin (TA), flavonoids (FL), total phenols (TP), chlorophyll-a (CA), chlorophyll-b (CB), and carotenoid (CR). Blue arrows represent the R, G, B color values corresponding to different body regions of the larvae: the head (R1, G1, B1), dorsal thorax and abdomen (R2, G2, B2), stripe (R3, G3, B3), dorsal mid-line (R4, G4, B4), and tail (R5, G5, B5). Axis-1 accounts for 35.57% of the variance, indicating the primary differentiation between substances and color indices, while Axis-2 represents 15.00% of the variance, offering insights into additional differentiations. The length and direction of the arrows reflect the strength and directionality of the correlation between the substances and the color indices.

**Table 1 insects-16-00728-t001:** Analysis of variance (three-way ANOVA) for main effects (age, host, and body region) on L (lightness) value, and their interactions.

Source	df	F	*p*
A	5	3134.899	<0.001
H	3	20.564	<0.001
B	4	5990.234	<0.001
A × H	15	12.027	<0.001
A × B	20	813.310	<0.001
H × B	12	13.674	<0.001
A × H × B	60	8.533	<0.001
Error	3390	—	—
Total	3510	—	—
Corrected total	3509	—	—

A: age; H: host; B: body region.

**Table 2 insects-16-00728-t002:** Interpretation rate of endogenous substance of host leaves to body color indices and Mote Carlo test results.

Endogenous Substance	Explain the Amount of Variation	Contribution/%	Pseudo *F* Statistic	*p*
SS	1.9	3.4	0.5	0.702
SP	5.1	8.9	1.3	0.270
TA	5	8.7	1.4	0.242
FL	9.1	15.9	2.1	0.086
TP	2.8	5	0.8	0.574
CA	8.2	14.3	2.1	0.074
CB	19	33.2	4.2	0.008
CR	6	10.6	1.5	0.218

SS: soluble sugar; SP: soluble protein; TA: tannin; FL: flavonoids; TP: total phenols; CA: chlorophyll-a; CB: chlorophyll-b; CR: carotenoid.

**Table 3 insects-16-00728-t003:** Regression of body color value of 20 day-old *Perina nuda* larvae with host color.

Body Part	Regression Model	*VIF_max_*	*F*	*p*	*R* ^2^
Head	Y_Rb_ = −9.640 + 0.159 Rh + 0.469 Gh − 0.076 Bh	9.314	4.215	0.008	0.138
	Y_Gb_ = 6.932 + 0.216 Rh + 0.192 Gh − 0.074 Bh	9.314	3.481	0.020	0.117
	Y_Bb_ = 12.953 + 0.095 Rh + 0.094 Gh − 0.001 Bh	9.314	2.159	0.099	0.076
Dorsal thorax and abdomen	Y_Rb_ = 84.31 + 0.329 Rh + 0.081 Gh − 0.271 Bh	9.314	0.818	0.488	0.030
	Y_Gb_ = 88.762 + 0.373 Rh − 0.135 Gh − 0.336 Bh	9.314	0.979	0.407	0.036
	Y_Bb_ = 86.007 + 0.414 Rh − 0.365 Gh − 0.218 Bh	9.314	0.905	0.442	0.033
Stripe	Y_Rb_ = −13.727 − 0.426 Rh + 0.649 Gh + 0.051 Bh	9.314	2.091	0.108	0.074
	Y_Gb_ = −6.752 − 0.396 Rh + 0.558 Gh + 0.028 Bh	9.314	1.539	0.211	0.055
	Y_Bb_ = −10.974 − 0.402 Rh + 0.527 Gh + 0.074 Bh	9.314	1.755	0.163	0.062
Dorsal mid-line	Y_Rb_ = 148.010 − 0.060 Rh + 0.052 Gh + 0.100 Bh	9.314	0.155	0.951	0.004
	Y_Gb_ = 171.122 − 0.009 Rh − 0.358 Gh + 0.050 Bh	9.314	0.875	0.458	0.032
	Y_Bb_ = 125.068 − 0.091 Rh − 0.561 Gh + 0.288 Bh	9.314	1.657	0.183	0.059
Tail	Y_Rb_ = 104.703 + 0.575 Rh − 0.627 Gh − 0.181 Bh	9.314	2.092	0.108	0.074
	Y_Gb_ = 15.556 + 0.150 Rh − 0.094 Gh − 0.004 Bh	9.314	1.076	0.364	0.039
	Y_Bb_ = −5.829 − 0.179 Rh + 0.205 Gh + 0.091 Bh	9.314	4.789	0.004	0.154

Rh, Gh, and Bh are the R (red), G (green), and B (blue) values of different host leaves, respectively. Rb, Gb, and Bb are the R, G, and B of the body coloration, respectively.

## Data Availability

The original contributions presented in this study are included in the article/Appendix A. Further inquiries can be directed to the corresponding author.

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
