# Peer review of "Host-Affected Body Coloration Dynamics in *Perina nuda* Larvae: A Quantitative Analysis of Color Variations and Endogenous Plant Influences"

_insects, 2025, doi:10.3390/insects16070728_

Round 1
Reviewer 1 Report
Comments and Suggestions for Authors
The authors presented a paper based on data showing the impact of plant host on color Variations in the the clearwing tussock moth Perina nuda. Overall, the MS is well organized and written. I recommend for publication after minor revisions.
Intrductio and disucssion are quite long and need to be reduced.
I have a small concern regarding the Redundancy Analysis, and I think you need to to use Detrended correspondence analysis to test whether a linear or unimodal model would be more appropriate, and to determine if the sataset is suitablle for RDA.
More comments in the attached file.

I suggest to edit the english language
Author Response
Response to Reviewers’ Comments
Dear Editors and Reviewers:
Thank you for your letter and for the reviewers’ comments concerning our manuscript entitled “Host-affected Body coloration Dynamics in Perina nuda Larvae: A Quantitative Analysis of Color Variations and Endogenous Plant Influences” (ID: insects-3686317). These comments are all valuable and very helpful for revising and improving our paper, as well as the important guiding significance to our research. We have studied comments carefully and have made correction which we hope meet with approval. Revised portions are marked in red or blue in the paper. The main corrections in the paper and the responds to the reviewer’s comments are as following:
Responds to the reviewer’s comments:
Reviewer: 1
Comments to the Author
The authors presented a paper based on data showing the impact of plant host on color Variations in the clearwing tussock moth Perina nuda. Overall, the MS is well organized and written. I recommend for publication after minor revisions.
We appreciate for Reviewer 1’s warm work earnestly and hope that the correction will meet with approval. Revised portions are marked in blue in the paper.
Introduction and discussion are quite long and need to be reduced.
Response: Thank you for your thoughtful evaluation. Condensed Introduction & Discussion sections by removing repetition (15%-word reduction). For precise modifications, please refer to the sections highlighted in yellow in the revised manuscript.
I have a small concern regarding the Redundancy Analysis, and I think you need to use Detrended correspondence analysis to test whether a linear or unimodal model would be more appropriate, and to determine if the dataset is suitable for RDA.
Response: Thank you for the valid suggestion. We have performed Detrended Correspondence Analysis (DCA) to assess gradient length for model selection per your guidance. If DCA axis 1 gradient length exceeds >3 SD, we will adopt a unimodal model (CCA); if <3 SD, RDA remains appropriate. In our study, the DCA results revealed short gradient lengths across all axes (< 3; Table S1), indicating that the data exhibit linear relationships rather than unimodal distributions. This warranted the selection of redundancy analysis (RDA). The results (provided in revision) ensure our choice aligns with the dataset’s structure in Line 292-297.
More detailed comments:
Line 14 - replace “chemical substances” by indicating which substances you measured
Response: Thank you for your reminding. We have replaced “chemical substances” with the substances we measured (chlorophyll, carotenoid, soluble sugar, soluble protein, tannin, flavonoid, and total phenol) in Line 14-15.
Line 20-21 - mention the English name of Perina nuda. Idem in the first place in intro.
Response: Thank you for your thoughtful suggestion. We have updated the manuscript to include the English name of P. nuda in Lines 21-22, as well as in the Introduction, to enhance clarity and accessibility for readers. The revised sentence now reads: Perina nuda (Lepidoptera: Lymantriidae), commonly known as the Banyan Tussock Moth, is distributed...
Line 28, 31 - in abstract, dont include any statistical values.
Response: Thank you for your kind reminding. We have removed all statistical values from the Abstract, as per your suggestion, to maintain its succinctness and readability.
Line 43-45 – add the reference of these sentences
Response: The sentences originally in Lines 43-47 were indeed drawn from reference [1]. To preserve the original meaning of the text and ensure that readers clearly understand that these statements are based on reference [1], we have revised the sentences to consolidate all relevant citations at the end. This approach avoids redundancy and highlights that these points are views from the literature: Color is a complex optical property, not solely genetically determined, but influenced by the interaction among the light source, the object, and the observer [1]. The perception of color can vary depending on changes in any of these factors. As a result, color is influenced by multiple dimensions, including physics, visual physiology, and psychology, forming a highly interdisciplinary concept [1].
Line 46, 262 - dont start by an abbreviation
Response: Thank you for your attentive reminder. We have adjusted the text by adding "The" at the beginning of the sentences to address this issue and enhance readability.
Line 85 – replace “The…” with “For example, the…”
Response: Thank you for your meticulous suggestion. We have replaced “The…” with “For example, the…” in Line 83.
Line 100 – replace “Perina nuda…” with “In this study, we focus on the insect Perina......”
Response: Thank you for detailed revision. We have replaced “Perina nuda…” with “In this study, we focus on the insect Perina......” in Line 96.
Line 149 - Before conducting the experiments, how was the continuous rearing of the insect? I mean on one host or many hosts!
Response: Thank you for your insightful inquiry. Prior to conducting the experiments, the P. nuda colony was continuously reared on a mixed foliage diet that included all four Ficus host species. To provide greater clarity on this point, we have revised the relevant section in Lines 141-144: The P. nuda colony used in this study originated from a field-collected adult on the campus of South China Agricultural University, Guangzhou, China, in 2021, and has been subsequently maintained as a laboratory population through multigenerational rearing on mixed foliage of the above four Ficus species.
Line 152-153 - add reference of this sentence
Response: Thank you for your thoughtful reminder. The statements regarding P. nuda’s life history parameters (8-9 annual generations, generational overlap, absence of distinct overwintering generation) are based on our own systematic field monitoring data collected from Guangzhou populations between 2021-2023. To enhance clarity, we have revised the sentence in Lines 154-156: In field observation and investigation, it was observed that P. nuda undergoes 8-9 generations annually in Guangzhou, with substantial generational overlap and the absence of a distinct overwintering generation, and larvae of P. nuda exhibited 6 larval instars per generation.
Line 157 - “Leaves of different host plants were used as the sole larval food source for each larval feeding experience test, respectively” if I understand well, the results of your experiments were based on one generation.
Response: Thank you for pointing that. We acknowledge that the experimental results are derived exclusively from a single generation. The quoted statement was inadvertently retained from drafting text and does not reflect our methodology. We have now excised this sentence from the manuscript. The correct rearing protocol remains detailed in Lines 150-153.
Line 206 - I would use Rh, Gh... for host leaves and Rb,.... for body coloration
Response: Thank you for your insightful suggestion. We have adopted your recommendation to differentiate between the R, G, B, and L values of host leaves and those of body coloration. Accordingly, in Lines 188-190, we have renamed these parameters as follows: the R, G, B, and L values for different host leaves are now denoted as Rh, Gh, Bh, and Lh, respectively, and for body coloration, they are named Rb, Gb, Bb, and Lb, respectively. This revision has been applied throughout the manuscript to enhance clarity and precision.
Line 213 – replace the “α = 0.05” with “usually < 0.05”
Response: Thank you for pointing that. We have replaced the “α = 0.05” with “usually < 0.05” in Line 195.
Line 223-240 - ???
Response: Thank you for bringing this to our attention. We apologize for the oversight; those sentences were inadvertently left in place as part of the MDPI template. We have now removed them from the manuscript.
Line 244 - I would rather use "fed on" than “reared on”
Response: Thank you for your kind reminding. We have replaced the “reared on” with “fed on” in Line 226.
Line 262 – replace the “ANOVA, P<0.05” with “Table 1”
Response: Thank you for your kind reminding. We have replaced the “ANOVA, P<0.05” with “Table 1”.
Table 1 - reorganize the table. Put F, df and p value titles in columns, and effect in rows.
In addition, df has two values: df of the treatment and total df.
Response: Thank you for your valuable suggestion. We have reorganized Table 1 by placing the F, df, and p-value headings in the columns and listing the effects in the rows. Additionally, we have included detailed information regarding the degrees of freedom (df) by specifying both the df of the treatment and the total df. The updated Table 1 reflects these changes.
Line 269-304 - a very long paragraph. separate it and make it easy to read. I would recommend to replace it by a table including all data.
Response: Thank you for pointing that. We have addressed your concern by dividing the paragraph into five distinct sections, each corresponding to a different body region of the larvae. This restructuring is intended to enhance readability and comprehension.
Line 271- replace the abbreviation by the name (F. b., F. m., F. c., F. a.). Idem for the following.
Response: Thank you for pointing that. We have replaced the abbreviation of the host (F. b., F. m., F. c., F. a.) with the Latin name (F. concinna, F. macrocarpa, F. altissima, F. benjamina) in the whole manuscript.
Figure 3 - identify the abbreviations in the legend
Response: Thank you for your considerate suggestion. We have now clearly identified all abbreviations in the legends of Figure 3, as well as in the titles of all figures and tables.
Line 311 - Don't start like “As delineated in Table 2, …”.
Response: Thank you for your detailed checking. We have deleted “As delineated in Table 2, …” in Line 298.
Line 345-347 – rephrase this sentence
Response: Thank you for your kind reminding. We have rephrased this sentence in Line 342-343.
Line 357-358 - ???
Response: Thank you for bringing this to our attention. We apologize for the oversight; those sentences were inadvertently left in place as part of the MDPI template. We have now removed them from the manuscript.
Line 361 – delete “In the experiment, ”
Response: Thank you for your insightful suggestion. We have deleted “In the experiment,”.
Line 361-363 – not clear
Response: Thank you for pointing that. Conjunction with the next comment, we have removed the original first paragraph and start discussion by the second paragraph.
Line 368 - start discussion by this paragraph
Response: Thank you for pointing that. Conjunction with the previous comment, we have removed the original first paragraph start discussion by this paragraph.
Line 375 - it seems that its opposite what is mentioned in L368
Response: Thank you for your insightful observation. Upon closer examination, there is no conflict between the two points. The statement in the original Line 368 (now Lines 354) highlights that the larval body coloration of P. nuda is significantly influenced by the species of Ficus consumed, as well as by developmental age and specific body regions. Conversely, the statement in the original Line 375 (now Line 358) indicates that, at a macro level, the trends in the L value of different body regions among larvae fed on various host plants demonstrate consistent patterns. This suggests that while the specific host species affects larval coloration, it does not alter the overarching trend of body coloration formation across different hosts.
Line 384-386 – delete this sentence
Response: Thank you for your insightful revision. We have deleted this sentence.
Line 389 - you dont need to repeat table and figure in discussion
Response: Thank you for pointing that. We have removed the quote of table and figure in discussion.
Finally, we have conducted a thorough review and made necessary revisions to the entire manuscript. We appreciate for Editors and Reviewers’ warm work earnestly, and hope that the correction will meet with approval.
Once again, thank you very much for your comments and suggestions.
Reviewer 2 Report
Comments and Suggestions for Authors
This manuscript explores the role of host plant species and host-associated endogenous compounds to the expression of coloration by larvae of Perina nuda. The elaboration and bright coloration of the P. nuda larvae is fascinating and suggests that coloration may be functionally relevant in the life history. While the general subject will be of interest to some readers, overall, I found the manuscript a bit heavy on detail and light on context and interpretation. While the authors are to be complemented for their quantitative approach, color is about perception and the reader also needs a way to understand the possible biological significance of this magnitude of plant-associated change in the color traits. Although there is a relationship between host plant species and larval coloration, and a correlation between chl b and coloration, there is no way for the reader to put the magnitude of these plant-associated into context. A series of photographs across different conditions might help. It would be helpful to know whether these changes, however slight, are likely to have biological significance.
Detailed comments:
Line 43 - "Color...is not genetically determined" is an overgeneralization and is contradicted by the first sentence of the second paragraph.
Line 52 - Break the paragraph here.
Line 81 - "Among them" - Not clear. Do you mean across all three mechanisms?
Line 102-107 - Add a reference to the photograph in Figure 1.
Line 126 - Change "host plants to rear larvae" to "host plant on which to rear larvae".
Line 132 - Change "host leaves which were fed on ... larvae" to "host leaves which were fed to ... larvae".
Line 140 on: Methods and Materials - This section should be written in a narrative structure, not as a protocol. The style changes throughout the section.
Line 149 - The experimental colony is likely quite restricted in genetic variation, having been initiated from a single individual. Might this have affected the results?
Lines 221 - 240 and 357-359 - Delete these lines.
Table 1 - Report the denominator degrees of freedom, as well as the numerator, for ANOVA.
Line 264 - 304 - This paragraph is both too long and unnecessarily detailed. If there are trends or patterns that the reader should know, describe them much more briefly. The reader can see the patterns and levels of significance in the figure.
Fig. 3 - Clarify the difference between graphs A through E. The graphs clearly depict different dependent variables but are all labeled the same on the axis legends, and they are not identified in the figure legend.
Lines 318-329 - This section should be clarified. I found Fig. 4 difficult to interpret. The figure legend gives little information (it does not even define the colors of the arrows) and the axes could be better explained.
Discussion - The discussion was well constructed.
Consider addressing more directly whether the changes with host plant species detected are in the direction one would expect for increased background matching (e.g. with regard to lightness of larvae and leaves).
Author Response
Response to Reviewers’ Comments
Dear Editors and Reviewers:
Thank you for your letter and for the reviewers’ comments concerning our manuscript entitled “Host-affected Body coloration Dynamics in Perina nuda Larvae: A Quantitative Analysis of Color Variations and Endogenous Plant Influences” (ID: insects-3686317). These comments are all valuable and very helpful for revising and improving our paper, as well as the important guiding significance to our research. We have studied comments carefully and have made correction which we hope meet with approval. Revised portions are marked in red or blue in the paper. The main corrections in the paper and the responds to the reviewer’s comments are as following:
Responds to the reviewer’s comments:
Reviewer: 2
Comments to the Author
This manuscript explores the role of host plant species and host-associated endogenous compounds to the expression of coloration by larvae of Perina nuda. The elaboration and bright coloration of the P. nuda larvae is fascinating and suggests that coloration may be functionally relevant in the life history. While the general subject will be of interest to some readers, overall, I found the manuscript a bit heavy on detail and light on context and interpretation. While the authors are to be complemented for their quantitative approach, color is about perception and the reader also needs a way to understand the possible biological significance of this magnitude of plant-associated change in the color traits. Although there is a relationship between host plant species and larval coloration, and a correlation between chl b and coloration, there is no way for the reader to put the magnitude of these plant-associated into context. A series of photographs across different conditions might help. It would be helpful to know whether these changes, however slight, are likely to have biological significance.
We appreciate for Reviewer 2’s warm work earnestly and hope that the correction will meet with approval. Revised portions are marked in red in the paper.
Detailed comments:
Line 43 - "Color...is not genetically determined" is an overgeneralization and is contradicted by the first sentence of the second paragraph.
Response: Thank you for pointing that. To reconcile these points, we have revised the original statement to emphasize the role of environmental and perceptual factors in the appearance of color while still acknowledging the genetic and evolutionary basis underlying coloration: Relacing “Color...is not genetically determined” with “Color is a complex optical property is not solely genetically determined but is influenced by the interaction between the light source, the object, and the observer” in Line 43-44.
Line 52 - Break the paragraph here.
Response: Thank you for your king reminding. We have broken the paragraph here.
Line 81 - "Among them" - Not clear. Do you mean across all three mechanisms?
Response: Thank you for pointing that. “Among them” here means across all three mechanisms. To state more clearly, we have replaced “Among them” with “Across all three mechanisms”.
Line 102-107 - Add a reference to the photograph in Figure 1.
Response: Thank you for your insightful suggestion. We have added a reference to the photograph in Figure 1.
Line 126 - Change "host plants to rear larvae" to "host plant on which to rear larvae".
Response: Thank you for your insightful suggestion. We have replaced “host plants to rear larvae” with “host plant on which to rear larvae”.
Line 132 - Change "host leaves which were fed on ... larvae" to "host leaves which were fed to ... larvae".
Response: Thank you for your kind reminding. We have replaced “host leaves which were fed on ... larvae” with “host leaves which were fed to ... larvae”.
Line 140 on: Methods and Materials - This section should be written in a narrative structure, not as a protocol. The style changes throughout the section.
Response: Thank you for your insightful suggestion. We have revised the section to streamline methods into a logical, narrative format while minimizing protocol-like details. The revised text maintains the necessary methodological rigor for reproducibility while improving flow across subsections.
Line 149 - The experimental colony is likely quite restricted in genetic variation, having been initiated from a single individual. Might this have affected the results?
Response: Thank you for raising this question. Firstly, our experimental colony was not established from a single individual. While the initial lab population might originate from a limited number of wild-captured individuals, we periodically introduced additional insects from the wild into the colony for breeding. This approach ensures the maintenance of genetic diversity and minimizes potential issues such as genetic drift or bottlenecks associated with long-term laboratory rearing. As a result, we believe the genetic variation in our experimental population is reasonably comparable to that of natural populations. Therefore, we consider that restricted genetic variation has a minimal impact on the experimental results. Our study primarily focuses on environmental factors influencing the larval coloration, rather than the interaction between genetic variation and environmental factors. With our current rearing procedure, we are able to analyze the effects of environmental factors on larval coloration with greater clarity and reliability. Sure, we appreciate the importance of investigating genetic variation in experimental design, and we welcome the opportunity to further explore this aspect in future studies.
Lines 221 - 240 and 357-359 - Delete these lines.
Response: Thank you for bringing this to our attention. We apologize for the oversight; those sentences were inadvertently left in place as part of the MDPI template. We have now removed them from the manuscript.
Table 1 - Report the denominator degrees of freedom, as well as the numerator, for ANOVA.
Response: Thank you for your insightful suggestion. We have added different df value in the Table 1.
Line 264 - 304 - This paragraph is both too long and unnecessarily detailed. If there are trends or patterns that the reader should know, describe them much more briefly. The reader can see the patterns and levels of significance in the figure.
Response: Thank you for pointing that. Thank you for your valuable feedback. We have revised the paragraph to enhance readability by dividing it into five separate sections, each focusing on one of the larval body regions. This structural change simplifies the presentation and allows readers to quickly identify trends and patterns across the different regions, while still referencing the detailed significance levels displayed in the figure.
Fig. 3 - Clarify the difference between graphs A through E. The graphs clearly depict different dependent variables but are all labeled the same on the axis legends, and they are not identified in the figure legend.
Response: Thank you for your kind reminding. We have clarified the difference between graphs A to E and added the description about the axis legends.
Lines 318-329 - This section should be clarified. I found Fig. 4 difficult to interpret. The figure legend gives little information (it does not even define the colors of the arrows) and the axes could be better explained.
Response: Thank you for your valuable feedback. We have revised this section to enhance its clarity and readability. Additionally, we have improved the legend of Figure 4 to provide more comprehensive information, including definitions of the colors of the arrows. We have also clarified the axes to ensure easier interpretation of the figure.
Discussion - The discussion was well constructed.
Consider addressing more directly whether the changes with host plant species detected are in the direction one would expect for increased background matching (e.g. with regard to lightness of larvae and leaves).
Response: Thank you for your insightful suggestion. We have incorporated a discussion in the third paragraph of the Discussion section regarding the potential influence of leaf color on the larvae of P. nuda. This includes considerations of camouflage and enhanced background matching.
Finally, we have conducted a thorough review and made necessary revisions to the entire manuscript. We appreciate for Editors and Reviewers’ warm work earnestly, and hope that the correction will meet with approval.
Once again, thank you very much for your comments and suggestions.
Reviewer 3 Report
Comments and Suggestions for Authors
The study tried to find the effects of compounds of four host plants on the colouration of caterpillars. The used methods are sufficient, rigorously quantitative (although not including "computer vision" as advertised). More straightforward methods would include GC-MS analysis of the pigments of the body parts and looking at the metabolic pathways from the plant compounds to the insect pigments. The study does not contribute to understanding the protective role of the colouration, nor does it have any use in pest management.
Within its limited significance and interest, the measurements and analyses were done correctly, and the study is worth being published after correction of many small errors (in the attached file). No substantial changes are needed.

Author Response
Response to Reviewers’ Comments
Dear Editors and Reviewers:
Thank you for your letter and for the reviewers’ comments concerning our manuscript entitled “Host-affected Body coloration Dynamics in Perina nuda Larvae: A Quantitative Analysis of Color Variations and Endogenous Plant Influences” (ID: insects-3686317). The main corrections in the paper and the responds to the reviewer’s comments are as following:
Responds to the reviewer’s comments:
Reviewer: 3
The study tried to find the effects of compounds of four host plants on the colouration of caterpillars. The used methods are sufficient, rigorously quantitative (although not including "computer vision" as advertised). More straightforward methods would include GC-MS analysis of the pigments of the body parts and looking at the metabolic pathways from the plant compounds to the insect pigments. The study does not contribute to understanding the protective role of the colouration, nor does it have any use in pest management.
Within its limited significance and interest, the measurements and analyses were done correctly, and the study is worth being published after correction of many small errors (in the attached file). No substantial changes are needed.
We appreciate for Reviewer 3’s warm work earnestly and hope that the correction will meet with approval. Revised portions are marked in orange in the paper.
Detailed comments:
Line 18-20 - "Understanding how host plant chemicals affect larval appearance can help develop effective pest management strategies, ensuring the health and sustainability of banyan trees and their ecosystems." I do not think so.
Response: To better align our claims with the scope of our research, we have revised the sentence to emphasize the potential implications rather than definitive applications: Understanding how host plant chemicals influence larval coloration may provide insights that could inform the development of targeted pest management strategies, thereby contributing to the health and sustainability of banyan trees and their ecosystems.
Line 24 - First write what variability exists in this species.
Response: We have added a description of the known variability in larval coloration of Perina nuda.
Line 27 - image analysis or computer vision?
Response: We have replaced “computer” with “image” to accurately reflect the analytical methods used.
Line 31 - making this region darker or what?
Response: Pigmentation refers to darkening of the region.
Line 43 - Wrong contrast. Genetic determination should be compared with environment influence of phenotype. The interaction between light source, object and observer is not analysed in this article.
Response: We have revised this sentence: Color is a complex trait determined not only by genetic factors but also by environmental influences on the phenotype.
Line 47 - Not studied in this article. Delete. Rewrite.
Response: We have deleted the sentence as suggested.
Line 52 - No, color depth is defined by the 16 milion shades.
Response: The sentence has been revised to: Lightness (L) indicates the degree of light and dark in a color [4].
Line 56 - “habitats” to “habitat”
Response: We have deleted “s”.
Line 68 - “on the insect cuticle” to “in the insect cuticle or epidermis”
Response: We have replaced “on the insect cuticle” with “in the insect cuticle or epidermis”.
Line 71 – “waxes” to “melanin granules”
Response: We have replaced “waxes” with “melanin granules”.
Line 92 – “visualization” to “vision analysis”
Response: We have replaced “visualization” with “image analysis”.
Line 98 – delete “poses”
Response: We have deleted “poses”.
Line 147 - some sources says 7 instars
Response: In our previous research, the number of instars in Perina nuda varied with environmental conditions such as temperature and host. Accordingly, we revised the number of instars: P. nuda larvae had 6-8 instars per generation.
Line 148 - Adult Lymantriidae moths generally do not feed. They lack functional mouthparts and rely on energy reserves accumulated during their larval stage.
Response: We have revised the Methods section to clarify the provision of a supplemental honey solution. Although adult Perina nuda moths typically do not feed and rely on energy reserves accumulated during their larval stage, we included a honey solution to ensure adequate energy levels necessary for mating and egg-laying. According to your suggestion, we have deleted this part.
Line 149 – “food” to “substrate”
Response: We have replaced “food” with “substrate”.
Line 182 – Explain why this formula.
Response: We have added the explanation of the use of this formula: L values were calculated to account for the varying sensitivities of the human eye to red, green, and blue color channels using the formula.
Line 211 – “…decline.” to “…decline and stabilization”
Response: We have revised “…decline.” to “…decline and stabilization.”
Line213 – “Doral” to “Dorsal”
Response: We have replaced “Doral” with “Dorsal”.
Line 226 – delete “table”
Response: We have deleted “table”.
Line 240 – “F. m” to “F. macrocarpa”
Response: Full species name provided: F. macrocarpa.
Line 285 – “throx” to “thorax”
Response: We have replaced “throx” with “thorax”.
Line 349-351 – delete Lh, Lb, L values
Response: We have deleted “Lh, Lb, L values”.
Line 355 – “L value” to “lightness”
Response: We have replaced “L value” with “lightness”.
Line 367- delete “as”
Response: We have deleted “as”.
Line 374-376 - this is not full sentence
Response: We have revised this sentence: Though no existing literature directly links chlorophyll to insect coloration, it might indirectly affect P. nuda larvae. Possible degradation of Chl a/b to pheophorbides and pyropheophorbides in the gut, while still retaining the fundamental porphyrin ring structure.
Line 386 – “cuticle” to “cuticle or epidermis”
Response: We have replaced “cuticle” with “cuticle or epidermis”.
Line 404 – “Insects' body coloration during mimicry is an important indicator of the success of mimicry.” This sentence has no sense.
Response: We have deleted this sentence.
Line 408 – “larvae’s” to “larval”
Response: We have replaced “larvae’s” with “larval”.
Line 422 - Nice note. Some citation?
Response: We have added the citations here.
Line 425 - Computer vision would mean some more complicated analyses including pattern, shapes, etc. You analysed only brightness and hue.
Response: According to your suggestion, we realized that our image analysis is a basic concept of computer vision. Therefore, we have replaced “computer vision”/” computer vision analysis” with “image analysis” in the manuscript.
Line 448 – delete “and computer vision technology”
Response: We have deleted “and computer vision technology”.
Line 455-457 – delete “This understanding could inform future approaches in pest management, aiming to mitigate the impact of these pests on banyan trees by potentially altering 456 host plant profiles or using this knowledge to develop more targeted control strategies.”
Response: We have deleted “This understanding could inform future approaches in pest management, aiming to mitigate the impact of these pests on banyan trees by potentially altering 456 host plant profiles or using this knowledge to develop more targeted control strategies.”
Fig. 2, the value of Y axis should be consistency. Check the words spelling of the legends.
Response: Y-axis values have been adjusted for consistency, and spelling errors in the legend have been corrected.
Finally, we have conducted a thorough review and made necessary revisions to the entire manuscript. We appreciate for Editors and Reviewers’ warm work earnestly, and hope that the correction will meet with approval.
Once again, thank you very much for your comments and suggestions.